# Photodynamic Therapy-Based Dendritic Cell Vaccination Suited to Treat Peritoneal Mesothelioma

**DOI:** 10.3390/cancers12030545

**Published:** 2020-02-27

**Authors:** Natalia Trempolec, Bastien Doix, Charline Degavre, Davide Brusa, Caroline Bouzin, Olivier Riant, Olivier Feron

**Affiliations:** 1Pole of Pharmacology and Therapeutics (FATH), Institut de Recherche Expérimentale et Clinique (IREC), UCLouvain, 1200 Brussels, Belgium; natalia.trempolec@uclouvain.be (N.T.); bastien.doix@gmail.com (B.D.); charline.degavre@student.uclouvain.be (C.D.); 2Institut de Recherche Expérimentale et Clinique (IREC) Flow Cytometry Platform, UCLouvain, 1200 Brussels, Belgium; davide.brusa@uclouvain.be; 3Institut de Recherche Expérimentale et Clinique (IREC) 2IP, UCLouvain, 1200 Brussels, Belgium; caroline.bouzin@uclouvain.be; 4Institute of Condensed Matter and Nanosciences (IMCN), Molecular Chemistry, Materials and Catalysis, UCLouvain, 1348 Louvain-la-Neuve, Belgium; olivier.riant@uclouvain.be

**Keywords:** photodynamic therapy, immunogenic cell death, danger-associated molecular patterns (DAMPs), radiotherapy, vaccination

## Abstract

The potential of dendritic cell (DC)-based immunotherapy to treat cancer is, nowadays, well documented. Still, the clinical success of immune checkpoint inhibitors has dampened the interest in anticancer DC vaccination. For highly life-threatening tumors that are regarded as nonimmunogenic, such as mesothelioma, however, T helper 1 immunity-biased DC-based immunotherapy could still represent an attractive strategy. In this study, we took advantage of photodynamic therapy (PDT) to induce immunogenic cell death to generate mesothelioma cell lysates for DC priming and evaluated such a vaccine to treat peritoneal mesothelioma. We found that the white light in vitro activation of the photosensitizer OR141 led to mesothelioma cell death, together with the release of bona fide danger signals that promote DC maturation. The administration of a PDT-based DC vaccine to mice bearing peritoneal mesothelioma led to highly significant survival when compared with sham or control animals treated with anti-CTLA4 antibodies. This was further supported by a strong CD8^+^ and CD4^+^ T cell response, characterized by an increased proliferation, cytotoxic activities and the expression of activation markers, including interferon gamma (IFNγ). Moreover, the PDT-based DC vaccine led to a significant increase in IFNγ^+^ T cells infiltered within mesothelioma, as determined by flow cytometry and immunohistochemistry. Finally, in vivo tracking of intraperitoneally administered DCs led us to document rapid chemotaxis towards tumor-occupied lymphatics (vs. lipopolysaccharide (LPS)-treated DC). DCs pulsed with PDT-killed mesothelioma cells also exhibited a significant increase in CCR7 receptors, together with an intrinsic capacity to migrate towards the lymph nodes. Altogether, these results indicate that PDT-based DC vaccination is particularly suited to induce a potent immune response against peritoneal mesothelioma.

## 1. Introduction

Malignant mesothelioma (MM) is an aggressive tumor of mesothelial cells, mainly related to asbestos exposure, that is pleural but also peritoneal and pericardial [1]. Due to its long latency and late diagnosis, the overall survival of patients is approximately one year, with a median progression free survival of less than six months [2]. Current options for patients with MM consist of a multimodal regimen of chemotherapy, using platinum-based compounds and folate antagonists, combined with anti-vascular endothelial growth factor (VEGF) antibody bevacizumab, surgery and radiotherapy. Unfortunately, due to the rapid development of resistance, treatments most often result in short-term regression, rapidly followed by local relapse [3]. The emergence of immunotherapy has offered some hope to improve MM patient outcomes. However, recent clinical studies have tempered the enthusiasm for conventional immune checkpoint blockers (ICB) to treat MM patients. Single agent anti-CTLA4 tremelimumab failed to show any survival benefits over a placebo, not reaching the primary endpoint in the largest MM immunotherapy study to date (NCT01843374) [4]. Clinical trials using single agent anti-PD-1 (nivolumab or pembrolizumab) or anti-PDL1 (avelumab) in the second or third line setting have demonstrated modest response rates [5,6,7,8] and no randomized comparisons against placebo or second line chemotherapy are currently available.

Several parameters may account for the limited MM response towards ICB. First, MMs show a low mutation rate compared to other cancers [9,10], reducing the amounts of tumor-associated antigens and, thus, the immunogenic potential of MM. Second, MM patients have usually low numbers of tumor-infiltrating CD8^+^ T cells whose function is either moderately or severely suppressed [3,11]. Third, MMs are usually associated with an immunosuppressive microenvironment including the presence of interleukin-10 (IL-10), VEGF, prostaglandin E2 (PGE2) and transforming growth factor-beta (TGF-β) [12,13]. Fourth, MM patients were reported to present a decreased number of circulating DCs (relative to healthy age and gender-matched controls) associated with a reduced ability to process antigens and a reduced expression of costimulatory and major histocompatibility complex (MHC) molecules [14].

Based on the above observations, dendritic cell (DC) vaccination may, thus, be more appropriate than ICB for MM immunotherapy. Ex vivo pulsed DC may indeed overcome the problems of low immunogenicity and deficiency in antigen presentation [15]. Several small clinical trials have already explored this possibility using autologous [16] or allogenic [17] MM lysate-pulsed DCs. Recently, the optimization of DC immunotherapy was proposed to integrate the concept of immunogenic cell death; in particular, to induce a superior T helper 1 (TH1)-mediated immunity [18]. Garg and colleagues have, for instance, identified hypericin-based photodynamic therapy (PDT) as a very efficient modality to develop DC-based vaccines that significantly increase the survival of high grade glioma-bearing mice [19]. Our lab also recently provided evidence that a new, non-porphyrinic photosensitizer named OR141 could induce immunogenic cell death (ICD) and thereby promote an anticancer immune response upon photodynamic activation [20]. This form of cell death results from the high tropism of OR141 for the endoplasmic reticulum (ER) that leads to the induction of a major ER stress response, further exacerbated by the inactivation of proteasomal deubiquitinases [21]. The capacity of OR141 to induce danger-associated molecular patterns (DAMPs) while killing cancer cells (and, thus, releasing antigens) upon white light LED activation [20] makes very feasible the use of this class of photosensitizers (PS) to challenge DCs before reinjection for vaccination.

When administering in vitro pulsed DCs, the major challenge is actually the in vivo priming of T cells in secondary lymphoid tissues. The route of DC administration, therefore, represents a major criterion for success. We thus reasoned that, for a subset of mesothelioma that develops in the abdomen, specifically on the peritoneum (i.p.), the injection of a DC-based vaccine could represent a particularly well suited strategy, considering the large population of lymph nodes throughout the gastrointestinal track and the presence of the spleen and gut-associated lymphoid tissues in the abdominal cavity. In the current study, we tested OR141 as a single ICD inducer to boost the priming of naïve DC with PDT-killed mesothelioma cells and, consecutively, to treat peritoneal MM in mice. We showed that such a PDT-based vaccine outperforms the anti-CTLA4 antibody administration, leading to significant tumor growth delay and tumor-free survival. We showed that not only was DC maturation stimulated, but also their migratory potential towards the peritoneal lymph nodes, giving rise to an efficient anticancer vaccination effect, with a high intratumoral recruitment of functional CD8^+^ T cells and a strong CD4^+^ TH1 response.

## 2. Results

### 2.1. OR141 Is a Potent ICD Inducer in Mesothelioma Cells

OR141 belongs to a new class of photosensitizers (PS) with a highly favorable light vs. dark efficacy ratio and a capacity to work under low pO_2_, as encountered in many tumors [21] (Figure 1A). We have recently shown that the treatment of mouse skin cancer lesions and subcutaneously grafted squamous cell carcinoma with OR141 led to significant growth inhibitory effects upon local exposure to white light photoactivation [20,22]. While the use of PDT to treat mesothelioma in situ is complicated due to accessibility issues, in vitro PDT may still be exploited to support the development of a DC vaccine primed with mesothelioma cells killed by photoactivated OR141. We first examined the capacity of OR141 to induce immunogenic cell death (ICD) in mouse mesothelioma cancer cells Ab1 and Ab12. We found that from the dose of 1 µM, OR141 induced cell death, as documented by the flow cytometry analysis using propidium iodide (PI) and Annexin V (Figure 1B). To determine the immunogenic nature of this cell death process, we examined the generation of several prototypical DAMPs upon exposure to cell death-inducing OR141 doses. A net increase in the amounts of extracellular HMGB1 and Hsp90 was observed upon exposure to photoactivated OR141 (Figure 1C). Flow cytometry analysis also revealed the OR141 dose-dependent translocation of calreticulin at the plasma membrane (Figure 1D).

### 2.2. PDT-Based DC Vaccination Outperforms Anti-CTLA4 Immunotherapy to Inhibit Mesothelioma Growth

To explore the immunogenic potential of PDT to treat mesothelioma, we primed immature bone marrow-derived DC with mesothelioma cells killed by photoactivated OR141. Phagocytosis was proven, using 5-chloromethylfluorescein diacetate (CMFDA), to pre-label cancer cells (Figure 2A and Appendix A) and, more importantly, DC maturation was documented by the increased expression of co-stimulatory molecules CD40, CD80 and CD86 (Figure 2B–D and Appendix A). Next, we examined the therapeutic potential of DC vaccination in vivo. To do so, we used syngeneic BALB/c mice for the Ab1 mesothelioma cells that we injected intraperitoneally (i.p.); Ab1 mesothelioma cells were chosen for this experiment since the in vitro release of DAMPs such as HMGB1 was larger than that obtained from Ab12 cells (see Figure 1C). A PDT-based DC vaccine was administered weekly i.p. from day four post-tumor inoculation for a period of 3 weeks (see Figure 2E). The generation of the PDT-based DC vaccine was obtained by exposing DC to mesothelioma cells treated with the highest OR141 dose tested (i.e., 10 µM) to achieve complete cancer cell avitalization. As a control immunotherapy, we used anti-CTLA4 antibodies (see drug regimen in Figure 2E), a strategy known to block major inhibitory signals at the DC surface. We found that mice receiving the PDT-based DC vaccine exhibited a significant antitumor response (vs. sham group) (Figure 2F–H and quantification at days 10 and 20 in Figure 2I and Figure 2J, respectively). Anti-CTLA4 treatment induced an initial reduction in tumor burden, but tumors eventually relapsed (see Figure 2G and quantification in Figure 2I,J). PDT-based DC vaccination actually led to an increased overall survival with 78% mice alive at day 55 (vs. 30% upon CTLA4 treatment and 7% for sham) (Figure 2K).

### 2.3. DC Priming with PDT-Killed Mesothelioma Cells Strongly Activate CD8^+^ and CD4^+^ Cells In Vitro

Next, we aimed to characterize the T cell response following either PDT-based DC vaccination or anti-CTLA4 antibodies administration. Using splenocytes collected at day 55 (i.e., the end-point of the experiments in Figure 2), we found that, upon re-exposure to Ab1 mesothelioma cells, CD8^+^ lymphocyte expansion was significantly more pronounced when splenocytes were derived from the PDT-based DC vaccination group than from the anti-CTLA4-treated mice (Figure 3A; gating strategy presented in Appendix A). This was associated with a robust cytotoxic activity upon the exposure of the mesothelioma Ab1 cancer cells to the lymphocytes collected from the vaccination group (Figure 3B). The PDT-based DC vaccination also showed a strong increase in the IFN-γ positive CD8^+^ population in comparison to the sham and anti-CTLA4 conditions (Figure 3C and Appendix A). Levels of LAMP-1, known to be critical for granzyme secretion, were also increased the in CD8^+^ population derived from the DC vaccination group (Figure 3D and Appendix A).

To evaluate T helper1 (TH1)-mediated immunity, we also focused on the CD4^+^ population using the same strategy as above. This led us to document an increase in both proliferation (Figure 4A) and INF-γ production (Figure 4B) in the CD4^+^ population, derived from splenocytes collected from mice treated with the PDT-based vaccine (vs. sham and anti-CTLA4 mouse groups; see also the gating strategy in Appendix A). An increase in the surface expression of the co-stimulatory receptor OX40 in the PDT-based DC vaccination group further validated CD4^+^ activation, supporting the capacity of PDT to enhance the TH1-mediated immune response [23] (Figure 4C).

### 2.4. PDT-Based DC Vaccine Induces a Potent Antitumor T Cell Response In Vivo

To further evaluate the T cell response in the tumor, we next examined the effects of a single injection of DCs primed with PDT-killed mesothelioma Ab1 cells (i.e., PDT-based DC vaccine) in mesothelioma-bearing mice. Tumors were collected six days after DC vaccination and were either analyzed by flow cytometry or immunofluorescence to quantify lymphocyte infiltration. A significant increase in tumor CD8^+^ T lymphocytes, as detected by flow cytometry, was observed in the PDT-based DC vaccine group (vs. control group and LPS-treated DC vaccine group, termed DC vaccine here below) (Figure 5A and Appendix A). These results were confirmed in tumor sections with a significant increase in CD8^+^ T cell population (Figure 5B,C). Furthermore, we found that PDT-based DC vaccination led to a significant increase in perforin expression and IFN-γ in the tumor (vs. untreated and DC vaccine) (Figure 5D,E). We also examined immune stimulation at the systemic level by comparing splenocytes six days after a single dose of either vaccine and looked for IFN-γ production. We found that CD8^+^ cells collected from the PDT-based DC vaccinated mice exhibited a larger increase in INF-γ than T cells derived from control mice and mice treated with the DC vaccine (Figure 5F). Notably, as further evidence of the strong immune response induced by PDT-based DC vaccination, a net decrease in tumor size was already observable six days after vaccination (Figure 5G).

### 2.5. PDT-Based DC Vaccine Administration Favors an Efficient Migration of DC toward Mesothelioma-Occupied Lymphatics

The differential effects obtained with DC vaccines involving PDT, or not, led us to suspect that, besides an increase in maturation, DC migration could play a major role in the therapeutic effects observed with PDT-based DC vaccination. We, therefore, labeled DCs with VivoTrack 680 NIR to examine their fate upon i.p. injection in mesothelioma-bearing mice. The fast chemotaxis of DCs primed with OR141-killed mesothelioma cells towards luciferase-positive tumor areas was detected only 15 min after injection, whereas LPS-treated DCs mainly remained located next to the injection point (Figure 6A, with the injection point marked with an arrow and Figure 6B for quantification). To further understand the reasons for the highly preferential DC migration towards the tumor area, we next examined the exact location of mesothelioma in our mouse models (obtained upon i.p. injection of mesothelioma cells). We found that the development of tumors consistently occurred in a specific area of the peritoneum that surgical dissection identified as the omentum area (i.e., the folds of the peritoneum that support the viscera) (Figure 6C). Furthermore, the injection of IRDye^®^ 800CW PEG (Li-cor Biosciences, Bad Homburg vor der Höhe, Germany), used as a lymphatic imaging agent, revealed that part of the homing of luciferase-positive mesothelioma cells was actually within the mesenteric lymphatic system, including nodes (Figure 6D).

Next, we collected mesothelioma six days post-DC vaccination to evaluate the presence of CD11c^+^ DCs within the tumor mass. While short-term measurements had documented net dissimilarities in the DC migratory potential (Figure 6A,B), we did not observe a significant difference in the intratumor DC density between DC and PDT-based DC vaccine conditions (Figure 7A,B). We then reasoned that the observed increased in DC migration reported in Figure 6A could eventually reflect a more efficient migration towards lymphatics and lymph nodes. This hypothesis was supported by the CCR7 measurements that revealed a significant increase in the abundance of this receptor (a major actor in the DC migration to the lymphatics) on DCs exposed to PDT-killed mesothelioma cells Ab1 and Ab12 (vs. LPS-treated DC) (Figure 7C). We finally used a footpad assay to directly compare the migration of the two DC populations towards the lymph nodes (LN). This assay revealed an increased density of DCs in the draining popliteal LN three days after the subcutaneous injection of the PDT-based DC vaccine into the hind limb footpad (vs. LPS-treated DC vaccine) (Figure 7D,E).

## 3. Discussion

We provide here several layers of in vitro and in vivo evidence that PDT-based DC vaccination represents an attractive strategy to target peritoneal mesothelioma. First, in vitro, exposure to DAMPs (and antigens) released from mesothelioma cells killed by PDT promotes DC maturation. Second, PDT-based DC vaccination outperforms the capacity of anti-CTL4 antibodies to induce CD8^+^ and CD4^+^ T cell activation in mesothelioma-bearing mice. Third, the PDT-based DC vaccine gives rise to a more significant recruitment of activated T cells within mesothelioma than LPS-treated DC. Fourth, the higher motility potential of DCs primed with PDT-killed mesothelioma cells leads to a more significant migration, upon i.p. injection, towards the peritoneal lymphatics and lymph nodes that are, in part, occupied by mesothelioma. Altogether, these data support a model wherein the anticancer effects of PDT-based DC vaccination result from the increased capacity of DC not only to induce a potent antitumor CD8^+^ and TH1 response, but also to rapidly reach peritoneal lymphatic structures where mesothelioma cells are homing. The observed response is all the more relevant as the mesothelioma models used in this study are described as poorly immunogenic [24].

Boosting DC response may be of particular interest in mesothelioma patients. Cornwall and colleagues documented a defect in DC numbers and maturation in mesothelioma patients; the larger the reduction in circulating DC, the shorter the survival of mesothelioma patients [14]. In particular, these authors identified a reduced ability to process antigens and a reduced expression of costimulatory and MHC molecules relative to the healthy controls. Moreover, such dysfunction was found to be irreversible, as documented by the incapacity of monocyte-derived DC from MM patients to process antigens to the same levels as their healthy counterparts [14]. The use of exogenously primed DC may further counteract the known immunosuppressive environment of mesothelioma [12]. In particular, the so-called next-generation DC vaccines aim to generate a potent T cell response and also induce a superior CD4^+^ T helper 1 (TH1)-mediated immunity [25,26,27] that is responsible for the persistence of cytotoxic T lymphocytes (CTL). Garg and colleagues have previously reported that by submitting glioblastoma cells to immunogenic cell death induced an immunostimulatory shift in the brain from regulatory T cells to TH1/cytotoxic T lymphocyte/TH17 cells [28]. In our study exploring the immunogenic potential of PDT, besides the high intratumoral recruitment of functional CD8^+^ T cells, we could document a net increase in IFN-γ-positive CD4+ cells. These TH1 cells further exhibited high levels of OX40, a signal previously reported to provide the co-stimulatory second signal to CD8^+^ T cell and also to diminish the inhibitory effect of Treg [29]. Another improved capability of DCs ex vivo primed with PDT-killed mesothelioma cells is related to their trafficking potential. The ability of mature DCs to mount a proper immune response is known to directly correlate with their ability to migrate to lymph nodes [30]. In our study, we showed that the PDT protocol was associated with a larger upregulation of CCR7 than that induced by LPS (however, this is known as a potent inducer of the migratory capacity of DC [31]). The high surface expression of CCR7 is associated with a high migratory responsiveness to the lymphatic-produced chemokines CCL19 and CCL21 [32,33,34]. Remarkably, we could document the increased motility of DCs in the minutes following i.p. injection, with a rapid distribution away from the initial delivery spot (Figure 6A). Together with the increased migratory capacity determined in the hind limb footpad assay, our results support the role of DC migration in the observed potent immune response. The extent and speed necessary to reach the T cell areas of the lymph nodes are known factors that participate in the T cell response, because of more efficient T cell cross-priming but also limited half-life of DCs and/or the MHC-peptide complexes that they express [35,36]. Finally, it should also be stressed that LN involvement at presentation occurs in 25–50% of pleural MM patient cases and the presence of nodal disease is a poor prognostic factor [37]. Thus, although the peritoneal route of administration is likely to account for a large part of the DC vaccine effects reported in our study, this increased DC migration towards LN could also be particularly suited for pleural MM patients.

Several immunotherapeutic approaches that exploit the tumor-associated differentiation antigen mesothelin are currently being explored to treat mesothelioma patients, including an anti-mesothelin antibody conjugated with chemotherapy (or not) [38,39] and an immunotoxin-coupled anti-mesothelin antibody variable fragment [40]. While these mesothelin-targeting strategies are being explored, DCs represent an attractive alternative. Various sources of tumor antigens can be used to pulse DCs ex vivo, including peptides, DNA and mRNA, but also tumor cell lysates whose antigens are not identified. The latter strategy is thought to lead to a broader repertoire of antitumor T cells (in response to the increased tumor-associated antigens (TAA) frequency) and to offer patient-individualized treatments when autologous tumor lysates are used to pulse DC. This modality has already been applied to mesothelioma patients [16,41,42]. Although encouraging, the use of whole-tumor cell lysate pulsing of DCs faces logistical limitations. First, mesothelioma cells need to be obtained in sufficient amounts from tumors, but many mesothelioma patients with advanced diseases are not candidates for surgical debulking or resection. Second, the implementation of this technique requires logistics dedicated to the work under good manufacturing practice (GMP)-certified conditions which, although manageable, may represent an economic brake being applied to widespread development. The use of mesothelioma cell lines, combined with PDT, to induce immunogenic cell death may, thus, help to generalize DC-based vaccination protocols. Recently, Aerts and colleagues confirmed that the use of allogenic tumor lysate was feasible [17] and launched a multicenter, randomized, phase II/III study of dendritic cells loaded with allogeneic tumor cell lysate [43]. The use of PDT to kill different human mesothelioma cell lines, instead of more conventional freeze-thaw protocols, would represent a minor adaptation to such protocol with a potentially relevant gain in efficacy.

Finally, our work also suggests that PDT directly administered to mesothelioma patients could promote an endogenous DC-based immune response, thereby even possibly offering more chances (than when using allogenic cell lines) for this to be directed against patient-specific mesothelioma antigens. The need for light activation of the photosensitizers would restrict such approach to peri-surgical times, although the use of OR141, which may be photoactivated by white light, could actually offer several advantages. Among them are a lower cost than that for PS that requires specific laser activation and the possibility to deliver the needed illumination through a conventional fiber optic endoscope. More generally, the intrinsic ability of PDT to reduce tumor burden may decrease the level of tumor-induced immunosuppression. Thus, PDT could offer what is desired when combining chemotherapy with immune therapy; that is to say, a reduction in the tumor burden obtained via the former in order to reach a state of minimal residual disease that is more prone to be eliminated by the latter [44,45,46]. Interestingly, the application of PDT as a local adjuvant for a lung-sparing surgical procedure in MM has already shown immunogenic effects [47,48].

## 4. Materials and Methods

### 4.1. Mesothelioma Cell Culture and Treatments

The Ab1 and Ab12 mesothelioma cell lines were acquired in the last three years from ECACC, where they are regularly authenticated by short tandem repeat profiling. The cells were stored according to the supplier’s instructions and used within 6 months after the resuscitation of the frozen aliquots. The Ab1 and Ab12 cells were maintained in DMEM/F12 GlutaMAX (Thermo Fisher Scientific Corp., Waltham, MA, USA) supplemented with 10% FBS and antibiotics. The luciferase-expressing Ab1 cell line (Ab1-luc) was obtained upon infection with (firefly) luciferase-encoding lentivirus particles (Amsbio, Abingdon, UK), followed by selection in 2 µg/mL puromycin. For PDT treatment, Ab1 and Ab12 cells were first treated with indicated OR141 concentrations for 1 h in the dark, before washing with PBS and photoactivation with a daylight LED source (2.55 mW/cm^2^) for 1 h (9.18 J/cm^2^), as previously determined [20].

### 4.2. Mice

All the experiments involving mice received the approval of the University Ethics Committee (approval ID 2016/UCL/MD018) and were carried out according to National Animal Care regulations. BALB/CByJ mice were obtained from Charles River (Charles River, Saint-Germain-Nuelles, France). Tumor xenografts were initiated by injecting i.p. 1 × 10^5^ Ab-1 Luc cells in 6-week-old BALB/CByJ mice. Tumor formation and progression was evaluated by live bioluminescence imaging using PhotonIMAGER™ (Biospace Lab, Nesles-la-Vallée, France).

### 4.3. Dendritic Cells and Vaccination

Bone marrow-derived dendritic cells (BMDCs) were generated from BALB/CByJ mice as described previously [20]. At day seven post-isolation, BMDCs were either exposed to 10 µM OR141-killed mesothelioma cells (as described above) at a ratio of 1:1 or LPS (from *E. Coli*, 0.5 μg/mL). DC maturation was analyzed with antibodies against CD11c-BV421 (BD Biosciences, San Jose, CA, USA, 565452), MHCII (I-A/I-E)-APC (BD Pharm, San Jose, CA, USA, 565367), CD40-PE (BD Pharm, 553791), CD80-PE (eBioscience, San Diego, CA, USA, 12-0801), CD86-PE (eBioscience, 12-0862) or CCR7-PE (BioLegend, San Diego, CA, USA, 120105). Live–dead exclusion was achieved by staining with FVD eFluor780 (eBioscience, 65-0865-14). Flow cytometry analysis was performed on FACS Canto II and data were analyzed using FlowJo software. For mouse vaccination, 2 × 10^6^ DC (in 100 μL PBS) were injected i.p. three times at one-week intervals; in parallel, another group of mice was also injected i.p. with 100 µg anti-CTLA4 (CD152) antibody (Bio X Cell, West Lebanon, NH, USA).

### 4.4. DC Tracking In Vivo

Seven-week-old female BALB/CByJ (Charles River) were injected intraperitoneally with 1 × 10^5^ luciferase-expressing Ab1-luc cells. At day seven, DC stained with the VivoTrack 680 NIR Fluorescent Imaging Agent (Perkin Elmer, Waltham, MA, USA) were intraperitoneally injected (4 × 10^6^ DC per mouse). PhotonIMAGER (Biospace Lab) was used to detect bioluminescence and fluorescence (excitation 680 nm, emission 730 nm) in the peritoneal cavity, either in vivo or ex vivo, after animal sacrifice.

In some experiments, 10^6^-labeled DCs were injected into mouse hind footpads and, at day three, mice were euthanized and the popliteal lymph nodes (PLN) were isolated and imaged as above.

### 4.5. Splenocytes Isolation and Activation

The mice were euthanized and the spleen was surgically removed and placed in PBS. The single cell solution was obtained by passing the spleen through a 70 µm strainer. Splenocytes were separated by gradient centrifugation with Ficoll-Paque™ PLUS (VWR, Radnor, PA, USA) and red blood cells were removed using red blood cells (RBC) lysis buffer (eBioscience, 00-4333-57). Splenocytes were cryopreserved in FBS with 10% Dimethyl sulfoxide (DMSO) until functional experiments. For activation, splenocytes were co-cultured with Ab1 cells in a ratio 10:1. Staining was performed using IFNγ-PE antibody (BioLegend, 505807) and CD107-PE (LAMP-1) antibody (BioLegend, 121611) together with protein transport inhibition using Golgi Stop (BD Pharm, 554715). T cell proliferation was determined based on the dilution of CellTrace™ carboxyfluorescein succinimidyl ester (CFSE) dye (Thermo Fisher Scientific Corp.) and cytotoxic activity was measured using PrestoBlue™ Cell Viability Reagent (Thermo Fisher Scientific Corp.).

### 4.6. Phagocytosis Detection

DCs were loaded with OR141-killed cancer cells which were pre-labeled with CellTracker™ Green CMFDA Dye (Thermo Fisher Scientific Corp.). After 18 h incubation, DCs were collected and stained with CD11c-BV421 (BD Pharm, 565452) and propidium iodide (PI). Any instances of double positives for CD11c and CMFDA were analyzed on FACS Canto II.

### 4.7. In Vitro DC Migration

Mature DCs (5 × 10^4^ cells in 50 µL RPMI 1640 growth medium) were added into the upper chamber of 5µm Transwell (Corning, New York, NY, USA) 24-well cell culture inserts. The lower chambers were filled with RPMI containing 100 ng/mL mouse recombinant CCL21 (R&D Systems, Minneapolis, MN, USA, 457-6C-025). After 2 h, cells from the lower chambers were collected and stained with BV421-CD11c antibody (BD Pharm, 565452) before analysis on FACS Canto II.

### 4.8. Tumor Infiltrating Lymphocytes (TIL) Analysis

For immunofluorescence studies, tumors were fixed for 24 h in 4% paraformaldehyde (PFA), followed by overnight incubation in a 30% sucrose solution. Tumors were then snap-frozen in optimal cutting temperature compound (OCT) and 5 μm sections were obtained. For CD8 (Cell Signaling, Danvers, MA, USA, 98941) and CD11c (Cell Signaling, 97585S) staining, slides were fixed for 10 min in 4% PFA before antigen retrieval in citrate buffer (pH 5.7) with 0.5% Triton. After the addition of an AlexaFluor-488-conjugated secondary antibody (Invitrogen, Carlsbad, CA, USA, A11034), slides were scanned (3DHistech Pannoramic P250 Flash III, Budapest, Hungary, 20X) and the extent of CD8 and CD11c staining was quantified over the total tumor area using Visiopharm software. For TIL analysis by flow cytometry, tumors were mechanically dissociated and single cell suspension was obtained by passing them through a 70 µm strainer. Cells were stained with CD45-BV510 (BD Pharm, 513151), CD3-APC (BD Pharm, 553066), CD8a-FITC (BD Pharm, 553031) and CD4-BV421 (BD Pharm, 553066) for 15 min RT.

### 4.9. Cell Death and DAMPs Detection by Flow Cytometry

For cell death profiling, cells were treated with the indicated OR141 concentrations, trypsinized and washed with PBS. Cells were consecutively incubated with fluorescein isothiocyanate (FITC)-conjugated Annexin V (Immunostep, Salamanca, Spain, ANXVF-200T) and 1 μg/mL propidium iodide (PI, Sigma-Aldrich, St. Louis, MO, USA), according to manufacturer’s instructions. After 15 min incubation at room temperature in the dark, cells were analyzed by flow cytometry on FACSCanto II (BD Pharm) with a gating strategy, excluding debris and doublet cells. For calreticulin translocation, cells were treated as described above and gently scraped off the plate after 6 h. After staining with an anti-calreticulin antibody (Abcam, Cambridge, UK, ab22683), a secondary goat anti-mouse APC-coupled antibody was added for 15 min at RT. Cells were then counterstained with PI and analyzed on FACSCanto II.

### 4.10. Immunoblotting

For protein extraction from the supernatant (conditioned media), the trichloroacetic acid (TCA) precipitation method was used. Briefly, the cell culture medium was centrifuged to remove cell debris before incubation with 2% sodium deoxycholate (1/1000 *v*/*v*) for 30 min on ice. TCA was then added to a final concentration of 7.5% and incubated on ice for 20 min. Proteins were recovered by high speed centrifugation (15,000 *g* for 20 min at 4 °C) before two washing steps with ice-cold acetone and resuspension of the protein pellet in RIPA buffer. Immunoblotting was performed as previously described [20]. Hsp90 (BD Biosciences, 610419) and HMGB1 (Abcam, ab18256) antibodies were diluted at 1/1000 (*v*/*v*) and β-actin antibodies (Sigma-Aldrich, A5441) at 1/2500 (*v*/*v*) in a Tris-buffered saline solution with Tween (TBST) (Tris-buffered saline, 0.1% Tween 20) solution with 5% *w*/*v* Bovine serum albumin (BSA).

### 4.11. Quantitative Real-Time PCR

qPCR was performed as previously described using the following couple of 5′–3′primers, for PRF1: CGCATGTACAGTTTTCGCCT and TGGTAAGCATGCTCTGTGGA; IFNG: CGGCACAGTCATTGAAAGCC and TGTCACCATCCTTTTGCCAGT; GAPDH: TATGTCGTGGAGTCTACTGGTGTCT and GGCGGAGATGACCCTTTTGGCT.

### 4.12. Statistics

Data are expressed as mean ± SEM of at least three independent experiments. The statistical significance between experimental conditions was determined by Student’s *t*-test or one-way analysis of variance (ANOVA, Tuckey’s post-hoc test). All data were analyzed with GraphPad Prism 7.0 (San Diego, CA, USA).

## 5. Conclusions

Our study provides evidence that PDT-based DC vaccination induces the effective cross-priming of cytotoxic CD8^+^ T cells. The DC maturation obtained upon exposure to DAMPs and antigens released upon OR141-induced ICD is further associated with a strong tumor-specific TH1 CD4^+^ response and the increased migratory potential of DCs towards the lymph nodes. Thus, PDT has the potential to gather together the properties of DCs that are usually dispatched between subsets of them: the so-called CD1c^+^ DCs that are migratory cells and mainly recognized for activating CD4^+^ T cells and CD141^+^ DCs with an enhanced ability to perform antigen cross-presentation and the activation of CD8^+^ T cells [49].

## Figures and Tables

**Figure 1 cancers-12-00545-f001:**
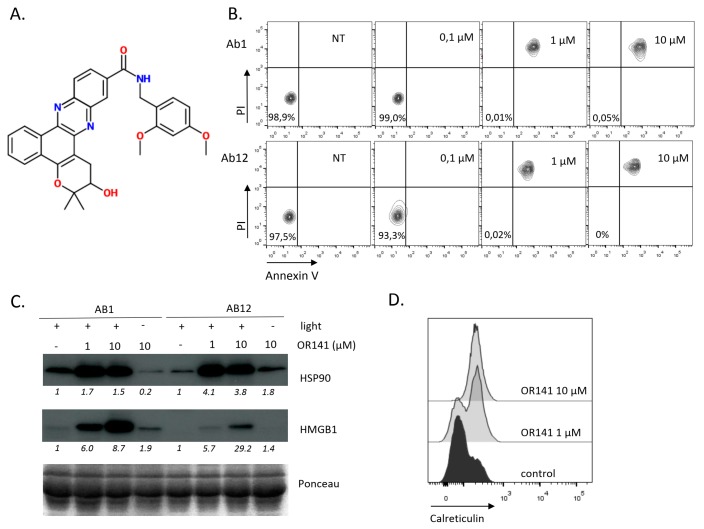
OR141 is an immunogenic cell death (ICD) inducer in mesothelioma cancer cells. Ab1 and Ab12 mesothelioma cells were treated with the indicated concentrations of OR141 for 1 h followed by exposure to white light LED for 1 h. (**A**) The molecular structure of OR141. (**B**) Cell death was measured using Annexin V and PI staining 18 h after OR141 photoactivation. (**C**) Extracellular (released) HSP90 and HMGB1 were detected by immunoblotting 18 h after OR141 photoactivation. (**D**) Calreticulin translocation in Ab1 cells 6 h after OR141 photoactivation was detected by flow cytometry. The data presented in this figure are representative of three different experiments.

**Figure 2 cancers-12-00545-f002:**
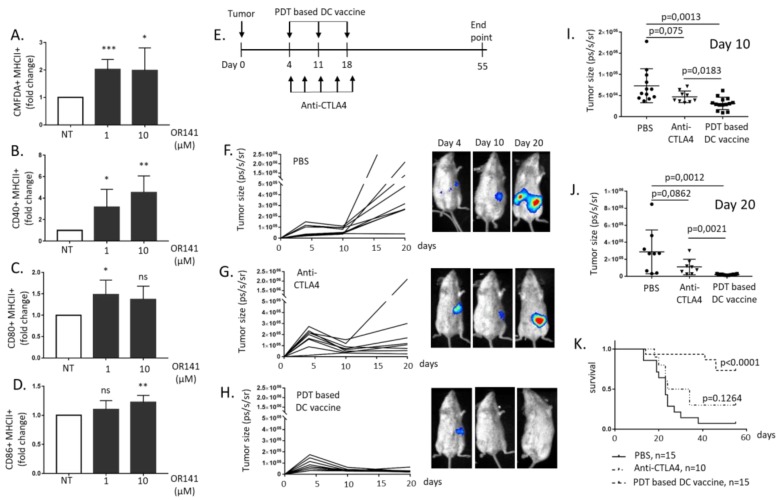
Photodynamic therapy (PDT)-based dendritic cell (DC) vaccination leads to potent mesothelioma growth inhibitory effects. Naïve DCs were exposed to lysates from OR141-killed Ab1 mesothelioma cells for 18 h. (**A**) The phagocytotic activity was evaluated based on the increased incorporation of CMFDA-labeled Ab1 in CD11c^+^ DC (identified by flow cytometry) (*n* = 3, * *p* < 0.05, *** *p* < 0.001). (**B**–**D**) The extent of CD40^+^/MHCII^+^ (**B**), CD80^+^/MHCII^+^ (**C**) and CD86^+^/MHCII^+^. (**D**) The DC populations, as determined by flow cytometry (*n* = 3, * *p* < 0.05, ** *p* < 0.01; ns, non-significant). (**E**) A schematic representation of the drug regimens used, namely PDT-based DC vaccination and anti-CTLA4 immunotherapy. (**F**–**K**) The BALB/c mice were injected with 10^5^ the luciferase-expressing Ab1 cell line (Ab1-luc) intraperitoneally (i.p.) and, at day four, were separated into three groups to evaluate the mesothelioma growth inhibitory effects of phosphate-buffered saline (PBS) (**F**), anti-CTLA4 antibodies (**G**) and the PDT-based DC vaccine (**H**); representative bioluminescence measurements are shown for each condition. The extent of the tumor size at day 10 (**I**) and day 20 (**J**) in the different conditions and corresponding Kaplan–Meier survival curves (**K**) (*n* = 10–15 mice).

**Figure 3 cancers-12-00545-f003:**
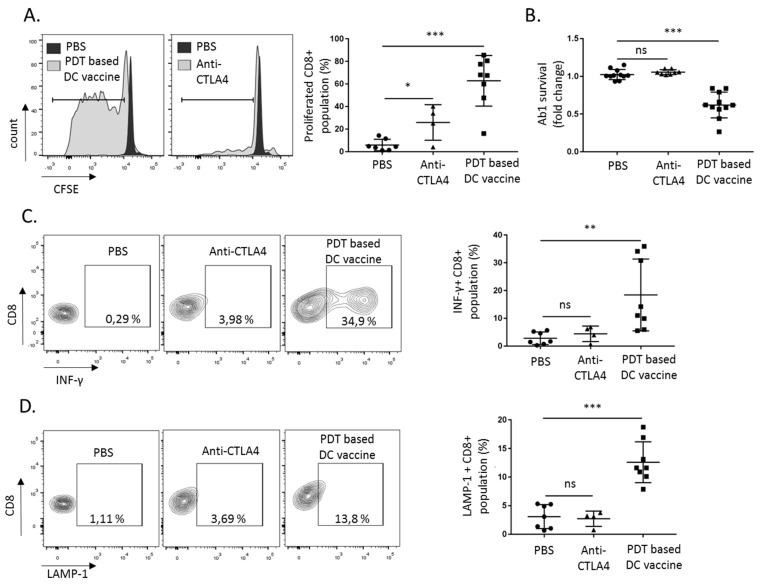
PDT-based DC vaccination promotes CD8^+^ T cell response. The CD8^+^ splenocytes collected at day 55 post-treatment (i.e., PDT-based DC vaccine or anti-CTLA4 antibodies, see Figure 2E) were re-exposed to the Ab1 mesothelioma cells. (**A**) The proliferation of CD8^+^ splenocytes was detected by CFSE dilution (*n* = 3, * *p* < 0.05, *** *p* < 0.001; ns = non-significant). (**B**) The survival of Ab1 mesothelioma cells, as determined four days after exposure to CD8^+^ cells (*n* = 3, *** *p* < 0.001; ns = non-significant). (**C**) IFN-γ production and (**D**) LAMP-1 staining were detected 18 h after Ab1 cell re-exposure (*n* = 3, ** *p* < 0.01, *** *p* < 0.001; ns = non-significant).

**Figure 4 cancers-12-00545-f004:**
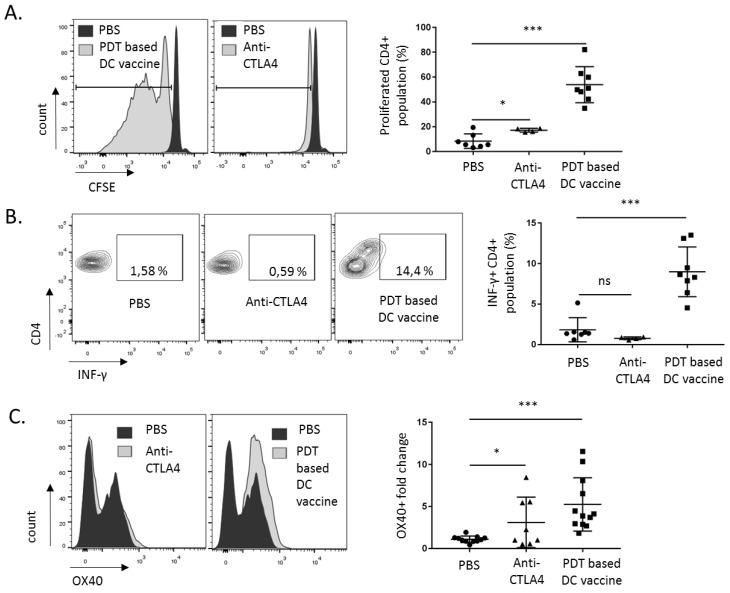
PDT-based DC vaccination promotes CD4^+^ T helper 1 (TH1) response. CD4^+^ splenocytes collected at day 55 post-treatment (i.e., PDT-based DC vaccine or anti-CTLA4 antibodies, see Figure 2E) were re-exposed to the Ab1 mesothelioma cells. (**A**) Proliferation of CD4^+^ splenocytes was detected by CFSE dilution (*n* = 3, * *p* < 0.05, *** *p* < 0.001; ns = non-significant). (**B**) IFN-γ production and (**C**) surface OX-40 staining were detected four days after Ab1 cell re-exposure (*n* = 3, * *p* < 0.05, *** *p* < 0.001; ns = non-significant).

**Figure 5 cancers-12-00545-f005:**
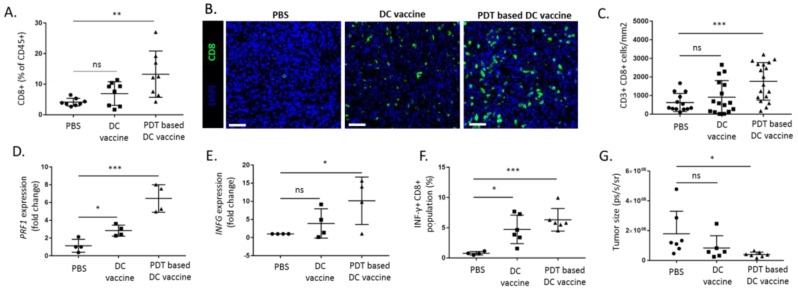
PDT-based vaccination induces intratumoral T cell response. BALB/c mice were injected with 10^5^ Ab1-luc i.p. At day six, mice were separated into three groups and treated i.p. with PBS, DC activated with LPS (DC vaccine) or PDT-based DC vaccine. Six days after single dose treatment, mice were sacrificed and tumors were collected for analysis. (**A**) Presence of intratumoral CD45^+^/CD3^+^/CD8^+^ cells was analyzed using flow cytometry (*n* = 8, ** *p* < 0.01; ns = non-significant). (**B**,**C**) The CD8^+^ immunostaining of indicated tumor sections, (**B**) representative images (scale bar = 50 µm) and (**C**) quantification (*n* = 15–20, *** *p* < 0.001; ns = non-significant). (**D**,**E**). The activity of tumor-infiltrating lymphocytes was analyzed based on the mRNA expression of *PRF1* (**D**) and *IFNG* (**E**) (*n* = 4, * *p* < 0.05, *** *p* < 0.001; ns = non-significant). (**F**) The detection of immunological response based on changes in INF-γ levels in antigens re-exposed CD8^+^ splenocytes using FACS (*n* = 4–6, * *p* < 0.05, *** *p* < 0.001; ns = non-significant). (**G**) The tumor burden was evaluated using a bioluminescence signal after one series of vaccination (*n* = 6–8, * *p* < 0.05; ns = non-significant).

**Figure 6 cancers-12-00545-f006:**
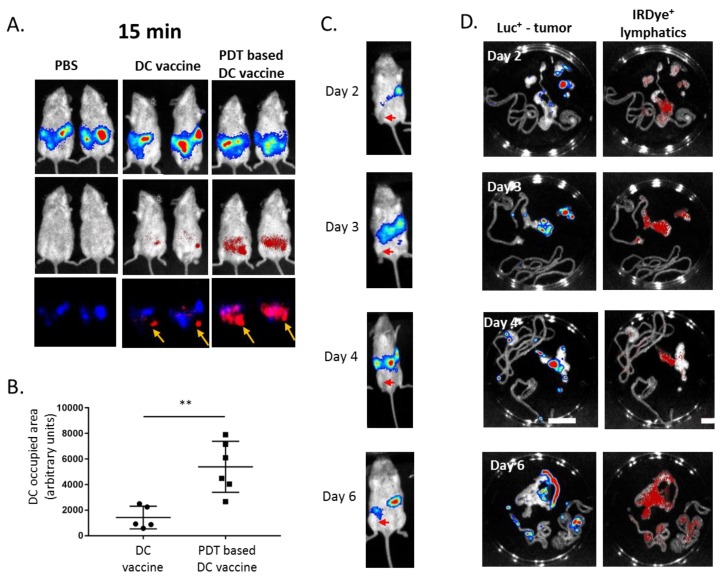
PDT protocol promotes the migration of DC towards tumor-occupied lymphatics. BALB/c mice were injected i.p. with 10^5^ Ab1-luc. At day six, mice were separated into three groups and injected i.p. with PBS, DC vaccine or PDT-based DC vaccine. Both DC populations were pre-labeled with VivoTrack 680 NIR and distribution was determined 15 min post-injection using in vivo fluorescence detection, together with the extent of the tumor burden evaluated by bioluminescence. (**A**) The representative images and (**B**) the quantification of DC occupied areas (*n* = 5, ** *p* < 0.01; ns = non-significant). (**C**) The time course of Ab1-luc tumor growth was monitored in vivo at the indicated time points; the arrows on the representative images indicate the injection site of luciferase-positive cells. (**D**) The Ab1-luc tumor-bearing mice were injected i.p. with IRDye^®^ 800CW PEG and sacrificed 30 min later. The representative bioluminescence (Ab1-luc) and fluorescence (lymphatics) signals from the intraperitoneal cavity, as revealed after rapid dissection.

**Figure 7 cancers-12-00545-f007:**
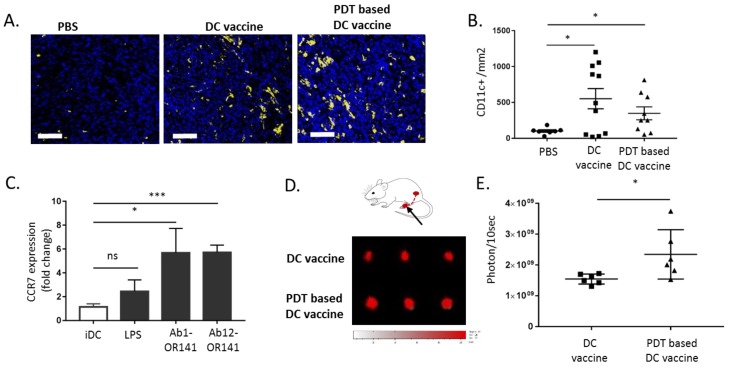
DC motility towards lymph nodes is increased upon priming with PDT-killed mesothelioma cells. (**A**,**B**) The tumor-bearing BALB/c mice were injected with PBS, DC vaccine or PDT-based vaccine and sacrificed six days later for tumor immunostaining. (**A**) The representative CD11c^+^ staining images (scale bar = 50 µm) and (**B**) quantification (* *p* < 0.05, *n* = 8–10). (**C**) The DCs were exposed to lysates from OR141-killed Ab1 and Ab12 cells or to LPS and the extent of CCR7 surface expression was determined by flow cytometry (* *p* < 0.05, *** *p* < 0.001, *n* = 3). (**D**,**E**) The migration of DC, pre-labeled with VivoTrack 680 NIR, from the mouse footpad to the popliteal lymph nodes. (**D**) A schematic representation and representative fluorescence pictures of popliteal lymph nodes (PLN) collected three days post-injection from mice treated either with the DC vaccine or PDT-based DC vaccine. (**E**) The quantification of the extent of fluorescence detected in the PLN (* *p* < 0.05, *n* = 6).

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
