# Peer review of "Photodynamic Therapy-Based Dendritic Cell Vaccination Suited to Treat Peritoneal Mesothelioma"

_cancers, 2020, doi:10.3390/cancers12030545_

Round 1

Reviewer 1 Report

Here the authors report dendritic cell vaccination for the treatment of mesothelioma. The vaccination was developed based on PDT using a photosensitizer named OR141. Upon OR141-PDT, mesothelioma cells were killed and released DAMPs, which subsequently promoted DC maturation and activated cytotoxic T cells. The encouraging results suggest that DC vaccination based on OR141-PDT has potential in mesothelioma treatment. Overall, this manuscript was clearly written, the topic is interesting and the results are supportive. Several issues that need to be addressed include:

Description of OR141 is vague. What’s the rationale of choosing OR141 vs other commonly used PSs? Why is OR141 favorable? Many other PSs have also been reported to trigger ICD.

What’s the absorption of OR141 and the fluence rate at this wavelength for PDT treatment?

Figure 1C. it’s more favorable if translocation is shown by fluorescence microscopy

Figure 6A, label each row.

Two cell lines (AB1 and AB12) were used for in vitro studies, and AB1 was used for in vivo experiments. Is there any reason why AB1 was selected instead of AB12?

What’s the subcellular localization of OR141 and what type of cell death is associated with OR141-PDT?

Author Response

We thank the Reviewer for the constructive comments.

Description of OR141 is vague. What’s the rationale of choosing OR141 vs other commonly used PSs? Why is OR141 favorable? Many other PSs have also been reported to trigger ICD. What’s the absorption of OR141 and the fluence rate at this wavelength for PDT treatment?

The text was edited (in the introduction, discussion and Methods, respectively) to read:

Our lab also recently provided evidence that a new non-porphyrinic photosensitizer named OR141 could induce immunogenic cell death (ICD) and thereby promotes anticancer immune response upon photodynamic activation (20). This form of cell death results from the high tropism of OR141 for the endoplasmic reticulum (ER) that leads to the induction of a major ER stress response further exacerbated by the inactivation of proteasomal deubiquitinases (21). The capacity of OR141 to induce danger-associated molecular patterns (DAMPs) while killing cancer cells (and thus releasing antigens) upon white light LED activation (20) makes very feasible the use of this PS to challenge DC before reinjection for vaccination.

The need of light activation of the photosensitizers would restrict such approach to peri-surgical times although the use of OR141 that may be photoactivated by white light could actually offer several advantages. Among them are a lower cost than that for PS requiring specific laser activation and the possibility to deliver the needed illumination through a conventional fiber optic endoscope.

For PDT treatment, AB1 and AB12 cells were first treated with indicated OR141 concentrations for 1h in dark before washing with PBS and photoactivation with a day-light LED source (2.55 mW/cm2) for 1 h (9.18 J/cm²) as previously determined (20).

Figure 1C. it’s more favorable if translocation is shown by fluorescence microscopy

We agree with the Reviewer that flow cytometry may be misguiding to detect calreticumin translocation since antibodies may stain intracellular calreticulin within dying cells. In our hands however, we counterstained cells with propidium iodide to exclude dying/dead (permeabilzed) cells and only analysed the right cell population exhibiting calreticulin translocation.

Figure 6A, label each row.

This figure panel was corrected.

Two cell lines (AB1 and AB12) were used for in vitro studies, and AB1 was used for in vivo experiments. Is there any reason why AB1 was selected instead of AB12?

The text was edited to read:

Ab1 mesothelioma cells were chosen for this experiment since the in vitro release of DAMPs such as HMGB1 was larger than that obtained from Ab12 cells (see Figure 1C).

What’s the subcellular localization of OR141 and what type of cell death is associated with OR141-PDT?

The text was edited to read:

Our lab also recently provided evidence that a new non-porphyrinic photosensitizer named OR141 could induce immunogenic cell death (ICD) and thereby promotes anticancer immune response upon photodynamic activation (20). This form of cell death results from the high tropism of OR141 for the endoplasmic reticulum (ER) that leads to the induction of a major ER stress response further exacerbated by the inactivation of proteasomal deubiquitinases (21).

Reviewer 2 Report

The manuscript sumitted by O. Feron and co-workers is of very high interest for Cancers. The work is properly performed, documented, and results are sound. The manuscript can therefore be published with very minor modifications. Figure 1, the structure of the photosensitizer should be recalled here. Line 74, explain abbreviation ICD. For the discussion line 318 I do not think GMP is a limitation of the technique. Indeed CAR T therapies are under clinics and the logistics are as complicated as here. Concerning the use of mesothelioma cell lines, the antigens released after PDT could be different from that of mesothelioma cells from the patient. Could authors discuss this point?

Thank you for hard and useful work.

Author Response

We thank the Reviewer for the constructive comments.

Figure 1, the structure of the photosensitizer should be recalled here.

Figure 1A now depicts OR141 molecular structure.

Line 74, explain abbreviation ICD.

The text was edited to read "immunogenic cell death".

For the discussion line 318 I do not think GMP is a limitation of the technique. Indeed CAR T therapies are under clinics and the logistics are as complicated as here.

We agree with the Reviewer that the procedure is certainly not more complicated than CAR T cell therapy but the anticipated cost remains elevated. We have now edited the text to read:

... implementation of this technique requires logistics dedicated to the work under Good Manufacturing Practice (GMP)-certified conditions which, although manageable, may represent an economic brake to a widespread development.

Concerning the use of mesothelioma cell lines, the antigens released after PDT could be different from that of mesothelioma cells from the patient. Could authors discuss this point?

We agree with the Reviewer and have now mentionned this issue in the discussion about the pros and the cons of allogeneic versus autologous approaches. The text has thus been edited to read:

Finally, our work also suggests that PDT directly administered to mesothelioma patients could promote an endogenous DC-based immune response, thereby even possibly offering more chances to be directed against patient-specific mesothelioma antigens (than when using allogenic cell lines).